# Challenges for Flood Risk Reduction in Poland's Changing Climate

Zbigniew W. Kundzewicz [1,*], Anna Januchta-Szostak [2], Elżbieta Nachlik [3], Iwona Pińskwar [1] and Janusz Zaleski [4]

1 Faculty of Environmental Engineering and Mechanical Engineering, Poznań University of Life Sciences, Piątkowska 94 E, 60-649 Poznan, Poland; iwona.pinskwar@up.poznan.pl

2 Faculty of Architecture, Poznań University of Technology, Jacka Rychlewskiego 2, 61-131 Poznań, Poland; anna.januchta-szostak@put.poznan.pl

3 Department of Geoengineering and Water Management, Cracow University of Technology, Warszawska 24, 31-155 Krakow, Poland; elzbieta.nachlik@gmail.com

4 Committee for Spatial Economy and Regional Planning, Polish Academy of Sciences, Pałac Kultury i Nauki, pl. Defilad 1, 00-901 Warsaw, Poland; janusz.zaleski@aspa.pl

* Correspondence: kundzewicz@yahoo.com

**Abstract:** Floods are the main natural disaster in Poland, and the risk of both fluvial and pluvial floods is serious in the country. Pluvial floods are on the rise in the changing climate, particularly in increasingly sealed urbanized areas. In this paper, we examine the changes in flood risk in Poland, discussing the mechanisms, observations, projections and variability. Next, we discuss flood risk management in the country, including specific issues related to urban and rural areas and the synergies between flood and drought risk reduction measures. We identify and assess the weaknesses of the existing flood risk management plans in Poland for the first planning period 2016–2021 and for the second planning period 2022–2027. We find the level of implementation of plans in the former period to be very low. Many planned measures do not have much to do with flood risk reduction but are often linked to other objectives, such as inland navigation. The plans contain numerous small measures, which come across as inapt and economically ineffective solutions. We specify policy-relevant recommendations for necessary and urgent actions, which, if undertaken, could considerably reduce flood risk. We also sketch the way ahead for flood risk management in Poland within the timeframe of the implementation of plans for 2022–2027 and the next regular update of plans for 2028–2033.

**Keywords:** natural hazards; flood risk reduction; climate change; adaptation; Poland





## 1. Introduction

The public awareness of water-related problems is not high in Poland, in spite of the considerable scientific interest in the issue. It is likely that many Polish people think of water when there is a spectacular problem related to water scarcity, its destructive abundance or inappropriate quality. Floods are directly related to the second of these challenges. However, problems with water quality may intensify when there is an excess of water; hence, distributed pollutants, including agricultural chemicals and nutrients, are flushed to the rivers. Flood-related water pollution can adversely affect environment quality and the state of ecosystems. Fluvial and pluvial floods have caused and continue to cause significant human, economic and social damage in Poland [1,2].

Since Poland is a member state of the European Union (EU), it is committed to implementing the regulations of the union. Therefore, efforts have been undertaken to harmonize the national legislation with EU directives. The two most relevant directives in the context of this paper are the Water Framework Directive [3] on the establishment of a framework for community action in the field of water policy—requiring EU member states to achieve

good status in all bodies of surface water and groundwater—and the Floods Directive [4], which directly deals with the assessment and management of flood risks.

In the Water Law [5] of Poland, flooding is defined similarly as in the EU Floods Directive [4], namely as a "temporary covering by water of an area that is not normally covered by water". However, inundations resulting from the overloading of urban drainage and sewage systems are excluded from the definition of the term "flood" in the Polish Water Law. Meanwhile, this type of water hazard is becoming more common and more severe in Poland.

Flood risk is often meant as a combination of the probability of a flood event and its adverse consequences. It is a function of three factors [6]: hazard (probability of occurrence of high river discharge, e.g., probability of crossing a threshold level), exposure and vulnerability. In the IPCC process [7], exposure is interpreted as the presence of people and valuable objects in places, which could be adversely affected, while vulnerability is defined as the predisposition or propensity to be adversely affected. There is a need for flood exposure analysis of critical and social infrastructure, so that a rational flood risk assessment can be performed [8,9]. In recent decades, the advantages of geospatial technologies and open-access data have made flood exposure analysis feasible at national scales, providing useful insights for policymakers and stakeholders.

Kron et al. [10] proposed a useful genetic classification of floods. In the summer, Poland features many convective storms with torrential rain, which may cause pluvial (flash and urban) floods. The largest fluvial (river) floods in the country also typically occur in the summer (June and July). They are born in the mountains and uplands in the south, where precipitation is high and floods are violent (Figure 1). Masses of water flow into two large river systems of the Vistula and the Odra, toward the Baltic Sea adjacent to the north of Poland. Cold-season floods also occur in Poland. A sudden thaw of abundant snow cover can lead to snowmelt flooding, and long cold spells in the winter can cause rivers to freeze, while warm weather in the winter or spring may cause the ice to break, leading to ice jams [11,12]. Gale-force winter storms may be accompanied by storm surges, which cause coastal floods, whose hazard is likely to grow with the sea-level rise accompanying the ongoing climate change.

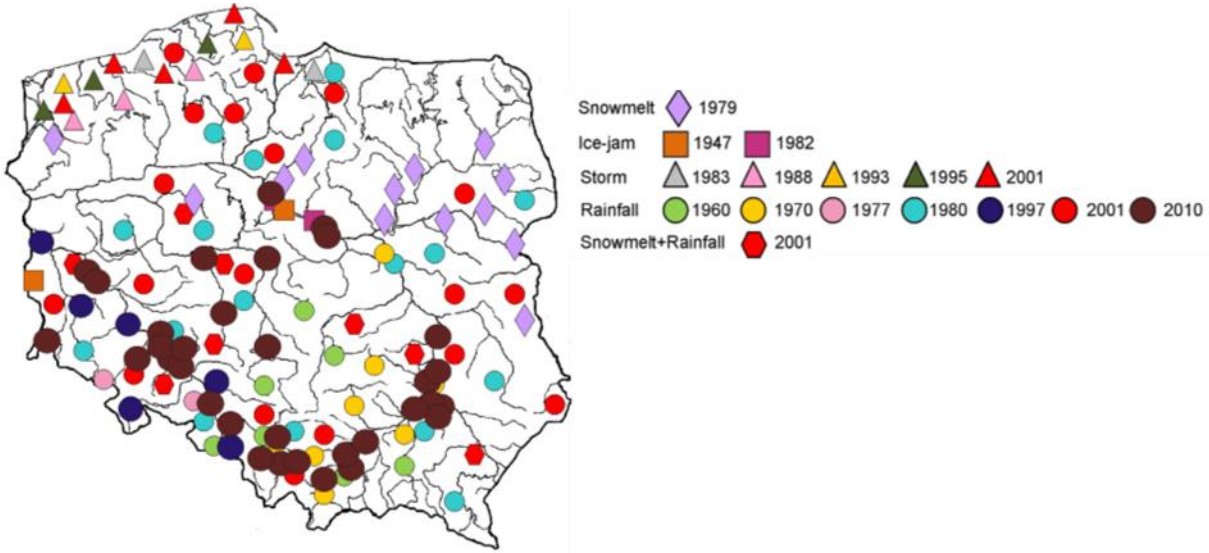

**Figure 1.** Spatial distribution of floods of regional extent in Poland since 1946. Concept modified according to Ref. [1]. The indicated symbols representing regional floods refer to the years 1947–2010. No large regional floods have occurred in Poland since 2010 to date (July 2023).

As proposed by Ref. [13], floods can be classified according to the mechanism of their occurrence. One mechanism consists of exceeding the protective capacity and design parameters (e.g., due to insufficient embankment height or insufficient flood polder capac-

ity). The other mechanism is a failure of the hydrotechnical infrastructure (e.g., structural damage to a dike or a dam, or a failure in the operation of pumping equipment or a flood gate). Finally, blockage of the capacity of the drainage system can occur.

The objective of this paper is to provide an overview of the state of flood risk management in Poland. The current legislation, regulations, plans, strategies and actions are referred to. Existing weaknesses are pointed out, and solutions and paths for improvement are proposed. The results are based on expert judgements and professional opinions of the co-authors, most of whom have gained long-term experience in various dimensions of flood risk reduction science and practice.

First, the changes in flood risk in Poland are discussed, and then, risk management is explored. The challenges of pluvial (flash and urban) floods are examined, whose frequency and intensity are on the rise. The synergies and conflicts between flood and drought risk reduction measures are also tackled, and relevant issues relating to water storage are discussed.

## 2. Flood Risk and Its Management in Poland

### 2.1. Changes in Flood Risk in Poland

The information on past floods in the Polish lands covers many centuries [1]. Large devastating river floods, which had hit the present Polish lands in the last three centuries, date back to August 1813, July 1903, July 1934, the summer of 1980, July 1997, July 2001 and May–June 2010 [1,14,15]. The most recent large snowmelt flood was recorded in March and April of 1979, while in January 1982, a destructive ice-jam flood was observed. There have been numerous storm surges and coastal floods along the Polish southern coast of the Baltic Sea.

It is fair to state, however, that Poland was not affected by other large floods, which devastated neighboring countries, e.g., in 2002 (Czech Republic and Germany), 2013 and 2021 (both in Germany). In August 2002, record high precipitation fell in the Upper Elbe (Labe) River Basin in the border area of the Czech Republic and Germany, and all-time national highs of 24 h precipitation were recorded in both countries—in Cinovec (Czech Republic) and Zinnwald (Germany) (cf., Ref. [16]). However, the precipitation over the Upper Odra River Basin, directly adjacent to the Upper Elbe Basin, was not that high and did not lead to major flooding in Poland.

Three catastrophic river floods in Poland in the last 30 years resulted in enormous economic losses (PLN 12.5 billion in 1997, PLN 3 billion in 2001 and PLN 12.5 billion in 2010, approximately equal to USD 3.6 billion, USD 0.7 billion and USD 3.7 billion, respectively), which were significant in relation to the GDP: 2.4% in 1997, 0.38% in 2001 and 0.9% in 2010 [17].

The devastating flood in July 1997 in the Odra River Basin was the most destructive natural disaster in Poland on record, as the losses caused by the event mounted to an all-time record high share of the Polish GDP. The flood was caused by a long wave of intense precipitation spells covering a large area of the Upper Odra River Basin in the Czech Republic and Poland on 4–10 July (after a rainy second half of June, overwhelming soil water capacity) (cf., Ref. [18]). At two stations in the Upper Odra Basin in Poland, the 5-day precipitation total exceeded 400 mm, while in the Czech part of the basin, it was even higher. The flood started in the Czech Republic, with the amplitude of stage and discharge exceeding anything recorded in the 20th century. In Poland, the town of Kłodzko on the River Nysa Kłodzka (Odra's tributary) was hit by extreme precipitation and flash flooding, with fatalities. On the Racibórz-Miedonia gauge on the Odra, the 100-year discharge of 1680 $m^3$/s was nearly doubled [19]. In Wrocław, where approximately a third of the city area was flooded, the maximum discharge reached 3600 $m^3$/s, while the flood protection system could accommodate 2400 $m^3$/s. The number of fatalities caused by the Odra flood in Poland reached 55. Over 100,000 people were evacuated. Nearly 50,000 houses and over 400 km of levees were inundated. The number of damaged road bridges was 198 on national roads and 1695 on provincial roads. Flood waters inundated

71 hospitals, 937 schools and kindergartens, 70 sewage treatment plants and 7 municipal solid waste repositories [18,20].

Figure 1 illustrates the spatial distribution of the occurrence of floods of regional extent in Poland since 1946. There have been no regional floods in Poland since 2010 (yet, many flash and urban floods have occurred virtually every summer). Most regional floods occurred in the catchments of the Vistula, the Odra and the coastal rivers. Many floods of regional extent occurred in the Sudety Mountains (southwest), the Central Carpathian Mountains in the south and in the valleys of large rivers, especially at the confluence of major tributaries.

The ongoing climate change causes a ubiquitous temperature rise, clearly observed in Poland, as the principal first-order effect. The warmer the atmosphere, the more water vapor it can hold, and hence, the higher the potential for occurrence of heavy precipitation, consistent with the Clausius–Clapeyron law (cf., Ref. [2]). Analysis of the 99th percentile of 24 h precipitation in Poland indicated an increase in precipitation in a more recent warmer period, with sensitivity for daily precipitation between 5.26%/°C and 6.06%/°C [21], but the rate is likely to be higher for shorter time (e.g., hourly) precipitation. This could translate into a higher risk of pluvial and fluvial flooding. However, a recent study of trend detection in high river discharge [22] did not produce persuading and definitive results. Generally weak and statistically insignificant changes were found. However, the existence of a spatial divide was detected, with increases in high river discharge in the south of Poland and decreases prevailing in the north.

In the last two decades or so, abundant precipitation has been recorded at many stations in Poland. Refs. [23–25], as well as Ref. [26], inform of high precipitation causing inundations, respectively, in Gdańsk (120 mm in 2001 and 160 mm in 2016), in Elbląg (81 mm in 2017), in Jodłownik, Małopolska Voivodship (152 mm in 2020), as well as in Poznań and its neighborhood in 2021 (up to 136.9 mm). Most recently, in September 2022, extreme daily rainfall of 130.4 mm was recorded in Gorzów Wielkopolski (https://klimat.imgw.pl/pl/biuletyn-monitoring/#2022/09 (accessed on 8 August 2023). This value was over 68% higher than the historical daily maximum to date at this station (77.4 mm in 1977).

Łupikasza and Małarzewski [27] noted that, based on the observation records from 1966 to 2020, the precipitation phase reacted significantly to the current warming: increasing trends in precipitation were strong and widespread, while decreasing snowfall trends were detected. This can translate into a higher risk of pluvial flooding also during the winter.

The projections for the future indicate an increase in heavy precipitation at the continental scale in Europe [28]. Additionally, in Poland, extreme precipitation is projected to increase with the warming, and this leads to an increasing risk of pluvial (flash and urban) flooding [2]. However, the trend in the projections of fluvial (river) flooding is less consistent in Poland, as well as in eastern Europe [29]. Piniewski et al. [30] noted an increase in the spatially varied projections of the high flow indicator. The patterns of projected changes were quite similar for different combinations of future timeframes and emission scenarios (RCPs). No significant changes were projected for the vast majority of rivers in the southern belt of the Vistula and the Odra Basins, while the highest increase was projected in small and medium-sized lowland rivers in the inner part of the basins.

The climate variability track in flood-related indices in Poland is rather weak and limited to the North Atlantic Oscillation (NAO), influencing winter precipitation [31].

*2.2. Flood Risk Management in Poland*

Despite heavy expenditures on flood preparedness in Poland, flood risk has been increasing in many areas for a range of reasons. Flood hazard has increased due to climate change increase in impervious areas and inappropriate river management. Flood exposure has increased due to construction in flood risk areas, while vulnerability has increased due to the growing number of people and the value of economic assets located in flood risk zones.

Large floods, such as the 1997 Odra flood, generated an impulse for undertaking intensive actions to reduce the future flood risk in Poland [32]. Activities aimed at flood risk reduction in Poland in 1998–2022 were co-financed from the national budget, international financial institutions and EU funds. Three large projects were led by The World Bank:

- The "Emergency Flood Recovery Project" implemented in 1998–2006, with a budget of USD 524.45 million (The World Bank, Implementation Completion Report, Loan 4264-POL, 2006);
- The "Odra River Basin Flood Protection Project" implemented in 2007–2020, with a budget of USD 1.019 billion (The World Bank, Implementation Completion and Results Report, IBRD-74360, 2021);
- The "Odra-Vistula Flood Management Project" implemented in 2015–2023, with a budget of USD 1.317 billion (The World Bank, Project Appraisal Document, PAD1203, 2015).

A financial reserve was earmarked in the Polish national budget for the response to natural disasters, and from 1997 to 2009, nearly PLN 10 billion (appr. USD 2.7 billion) was allocated [33]. As a result, Poland has significantly improved its flood protection system. However, the results in the Odra River Basin were much better than in the Vistula River Basin, where they are still unsatisfactory. The multi-objective program "ODRA 2006" for the Odra Basin, whose main component was devoted to flood protection [34], was approved by the Polish parliament with appropriate financing in the year 2001, while the Governmental Program for Flood Protection in the Upper Vistula Basin was approved by the government much later, in the year 2011 [35]. However, despite all the measures undertaken since 1997, flood risk reduction in Poland still requires intensive actions, especially in the Vistula River Basin. In the whole of Poland, the risk reduction of flash flooding (in both highland and urban areas) is a serious challenge.

After the 1997 flood, the flood protection system in Poland, and in the Odra River Basin in particular, has been considerably strengthened. Nowadays, a discharge of the size of the 1997 maximum (3600 $m^3$/s) is likely to pass through the city of Wrocław without causing major damage.

It is well recognized that complete flood protection is not possible, but what one could and should aim for is reducing flood risk and flood damage to the people, economy, environment and cultural heritage.

Flood risk reduction and climate adaptation are policy objectives, and flood risk management is the process through which these objectives are pursued. The flood risk management system in Poland is based on structural and non-structural measures (cf., Ref. [36]). The measures taken so far (https://www.wody.gov.pl/nasze-dzialania/plany-zarzadzania-ryzykiem-powodziowym (accessed on 8 August 2023) to reduce the risk of river flooding are mostly based on structural solutions and include

(i) flood levees with a total length of over 8600 km, mainly in first- and second-order rivers;
(ii) flood water retention in storage reservoirs, which is one of the functions of multi-purpose reservoirs (about 100 reservoirs have a capacity greater than 1 million $m^3$) (cf., https://www.gov.pl/web/premier/projekt-uchwaly-rady-ministrow-w-sprawie-przyjecia-programu-przeciwdzialania-niedoborowi-wody-na-lata-2022-2027-z-perspektywa-do-roku-2030 (accessed on 8 August 2023);
(iii) dry reservoirs and river polders (of significantly lower storage value);
(iv) development of streams and rivers, as well as stabilization of their channels (mainly in the south of Poland in the Upper Vistula and Upper Odra Basins).

Figure 2 illustrates the location of large and medium-sized water storage reservoirs, as well as polders and dry reservoirs in Poland.

Non-structural measures embrace activities, which increase catchment and river retention; improve monitoring, forecasts and information on flood hazards and risk; and devise operational plans for response and recovery. They are related to the adaptation of spatial planning to reduce the existing risk and its future increase (https://www.wody.gov.pl/nasze-dzialania/plany-zarzadzania-ryzykiem-powodziowym (accessed on 8 August

2023)). The current update of flood risk management plans for 2022–2027 [37,38]—while maintaining technical measures and their reconstruction and modernization after a flood—develops a package of non-structural measures aimed at

(i)   protecting or increasing catchment retention in forested areas, in agricultural lands and in built-up and urbanized areas;
(ii)  development of local flood warning and response systems;
(iii) extensive support for affected communities, mainly related to economic, technical, organizational and health support.

However, the existing hydrological and flood protection design standards have to change in order to encompass climate change and variability, with the associated uncertainties [39], while the current water management practices may be inadequate for reducing the adverse impacts of climate change in the future [40]. There is a clear gap between the results of scientific studies and the needs of practitioners in the domain of climate change adjustments in the engineering design of the flood risk reduction system.

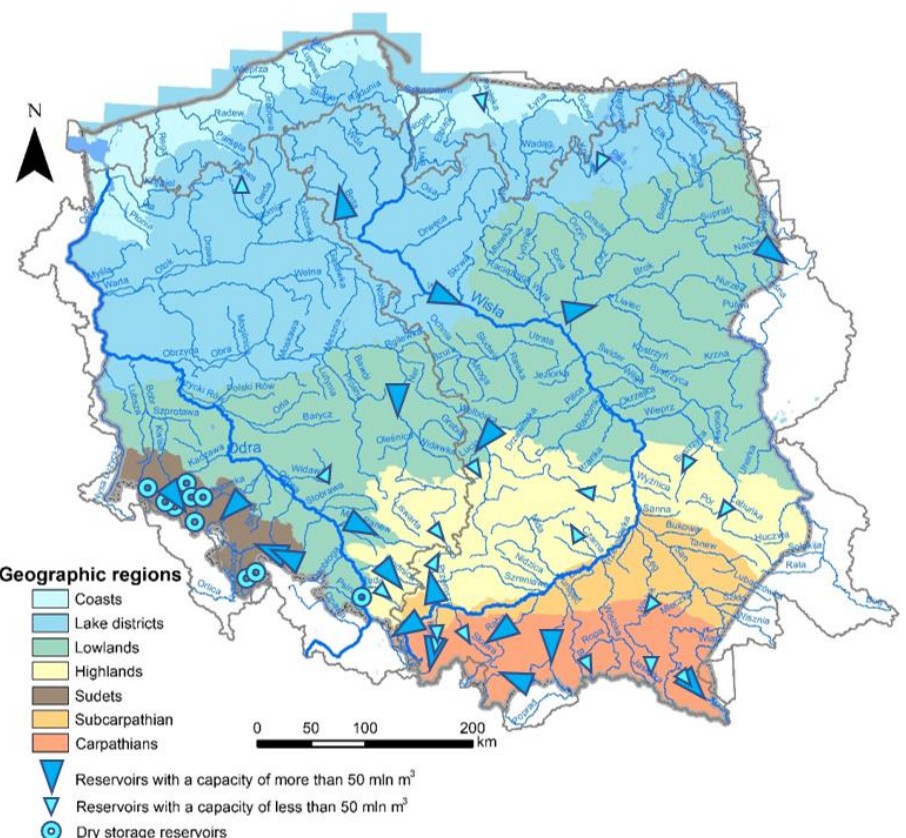

**Figure 2.** Spatial location of large, medium and dry storage reservoirs in Poland. The territory of Poland is colored. Parts of drainage basins, which extend beyond the territory of Poland (in Germany, the Czech Republic, Slovakia, Ukraine, Belarus, Lithuania, Russian Federation), are not colored. Based on Ref. [41] and revised.

Flood hazard is clearly non-stationary and changes over time [42]. The non-stationarity can be attributed not only to climate change and variability but also to land use and development (deforestation, urban sprawl, sealing of ground surface and changes in the channel). Therefore, the present estimate for a 100-year flood for a particular location can be dramatically different from (and often higher than) a 100-year flood determined for pre-development watersheds. That is, an event with discharge corresponding to historical 100-year flood is likely to recur more frequently in the future. In Poland, this problem is exacerbated by the aging of the flood protection infrastructure, mainly the retention reservoirs and levees [41]. Sedimentation of the bowl of multi-functional water storage

reservoirs causes their flood retention volume to decrease [43]. Due to the lack of the necessary modernization of the spillway systems, under the conditions of increasing maximum flows, their real effectiveness in reducing the flood wave does not correspond to the initial assumptions. This also applies to flood levees, which, with the increasing amplitude of river discharge, reduce the effectiveness of the operation; therefore, an assessment on a regional and supra-local scale in the country is required. Such assessment should provide the basis for undertaking appropriate modernization solutions for these facilities in conjunction with other complementary activities to increase the water storage capacity of the entire catchment rather than of individual facilities, which has been a common practice so far.

　　　Aside from attempts to reduce the water load via structural measures, one can try to enhance the resistance and resilience of the flood risk reduction system as a whole, as well as the society's readiness to live with floods. We may strive to build a resilient "safe-fail" system, which may occasionally fail in a "safe" way and is capable of bouncing back to a satisfactory state. Such capability is an essential property of resilience [44]. Unfortunately, we do not manage to either keep destructive waters away from people at all times (the essence of flood defense) or to keep people away from destructive waters at all times (the essence of flood prevention) (Figure 3). Therefore, it is necessary to embark on a diversified portfolio of flood risk management approaches, including flood risk mitigation, preparation and recovery, in order to maximize the net effect of a combination of strategies [44,45].

**Figure 3.** Flood risk reduction strategies. Based on Ref. [46] and modified.

　　　The Floods Directive [4] calls upon all EU member states to execute preliminary flood risk assessment, to prepare flood hazard maps and flood risk maps and to develop flood risk management plans. Flood hazard and flood risk maps for the whole territory of Poland have been developed for 0.2%, 1% and 10% events (cf., Ref. [47]), and they are available at the Hydroportal of the National Water Management Authority (KZGW) (https://mapy.isok.gov.pl (accessed on 8 August 2023)) (see example in Figure 4).

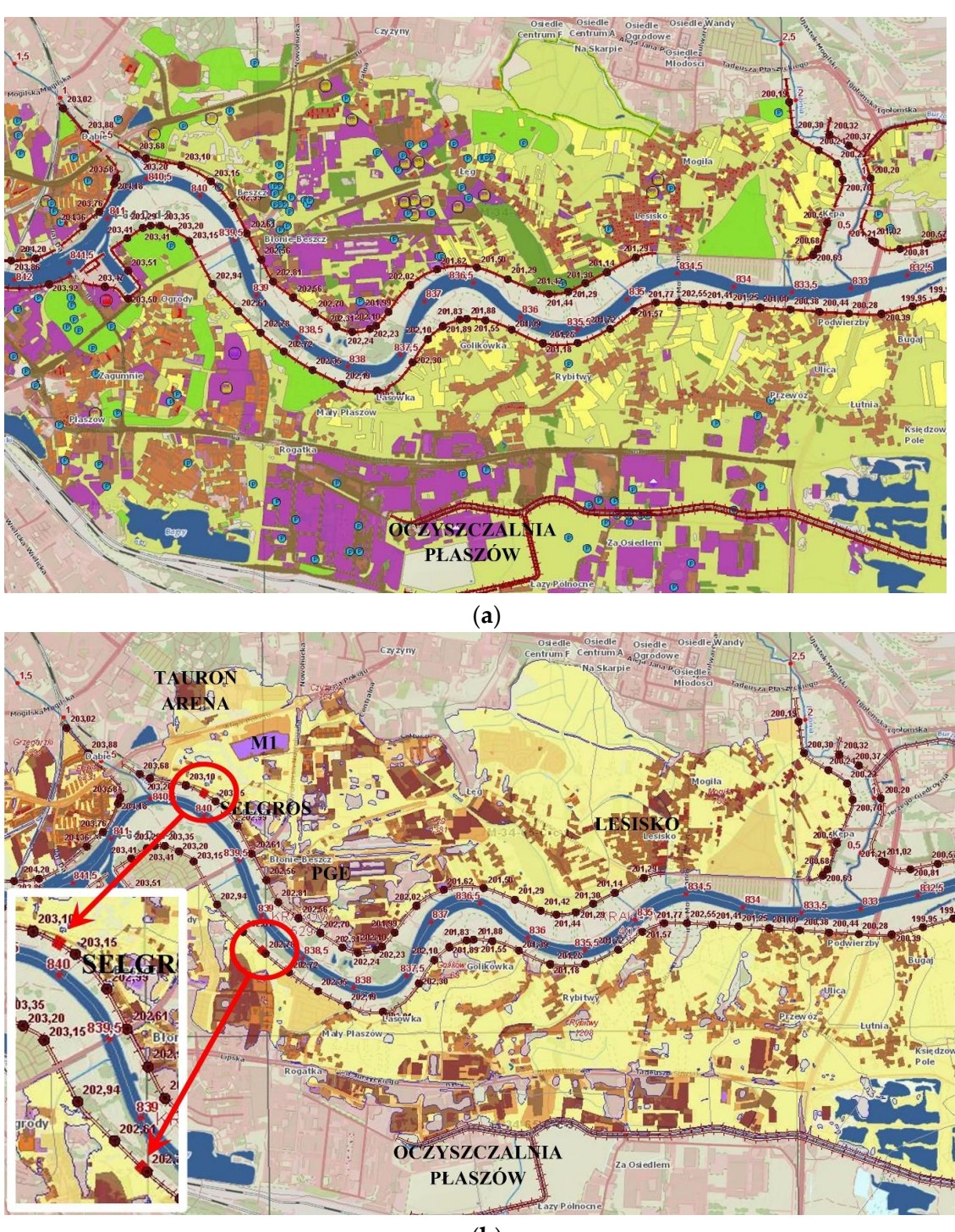

**Figure 4.** *Cont.*

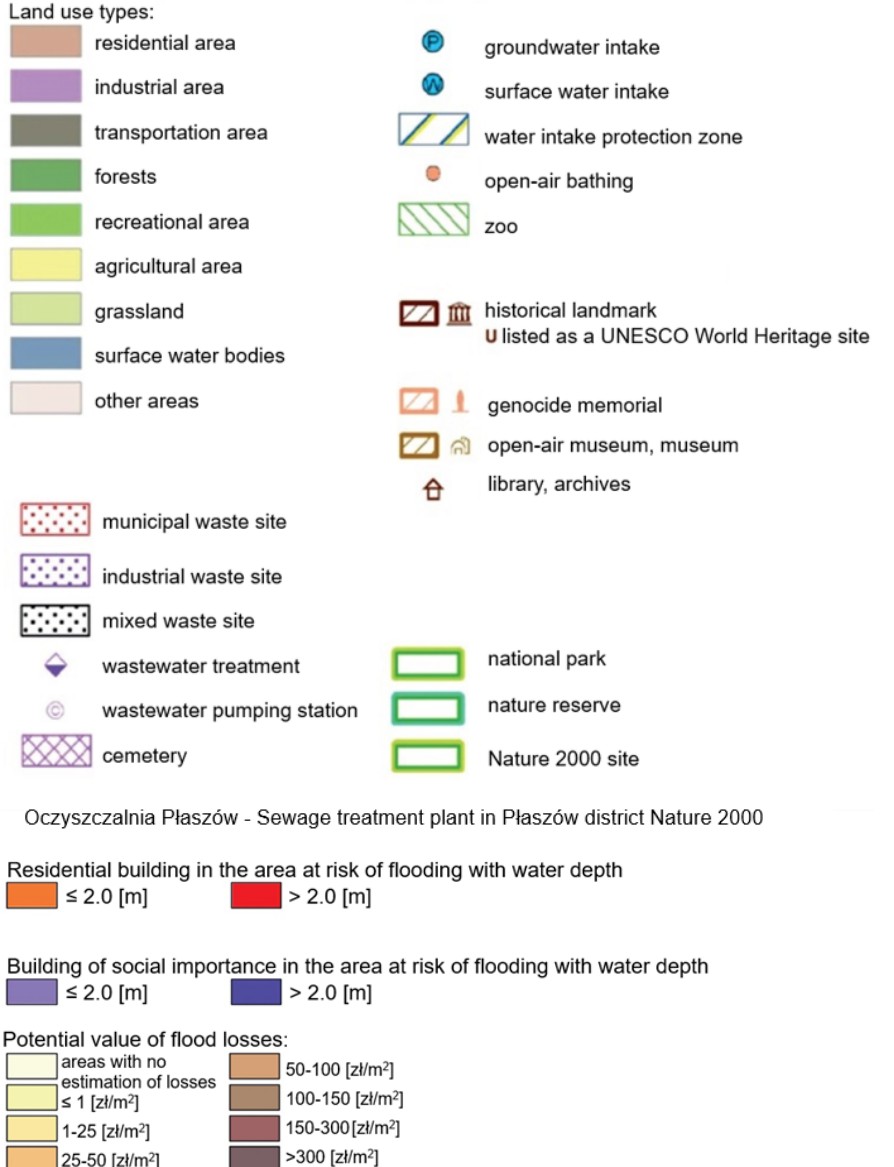

**Figure 4.** Exemplary flood risk map of Kraków for $Q_{1\%}$ (i.e., 1% flood probability—average recurrence once every 100 years) for two conditions: (**a**) dikes destroyed totally along both river banks; (**b**) dikes destroyed pointwise along both river banks (at points marked with red-filled rectangles). Source of map: Ref. [13], drawn from the Hydroportal of the National Water Management Authority (KZGW) (https://mapy.isok.gov.pl (accessed on 8 August 2023)).

Flood risk management plans (FRMPs) are strategic documents for river basins—obligatorily required by the EU Floods Directive—containing a set of measures, whose implementation should result in flood risk reduction. Plans should be subject to revision every six years. Unfortunately, the implementation of these documents has not been satisfactory thus far in Poland [48] (see Sections 3 and 4).

### 2.3. Challenge of Flash and Urban Floods

Heavy rainfall events have become more frequent and more intense in the warming climate of Poland, increasing the risk of pluvial floods [2,49]. Indeed, many severe pluvial (flash and urban) floods have been recorded in recent years in Poland (e.g., Refs. [24,26,49]), which could be attributed to both climatic and non-climatic factors. The latter include an increasing area of sealed ground surface, reduction in water storage in the catchments, a shortage of green spaces and—overall—the principle of fast

drainage of excess rainwater in cities via storm sewage systems [50]. Refs. [51,52] documented a number of examples of the adverse effects of "revitalization" processes in the country, consisting mainly of substituting natural green areas in towns with abundant impervious surfaces. This results in the overloading of urban drainage systems, localized inundations, as well as violent and frequent discharge of stormwater and untreated municipal sewage from the combined sewage systems, which cause increased pollution of surface waters and prevent the achievement of the WFD objectives. We are not prepared to address urban flood hazards nowadays or to reduce the effects of such floods in the near future. The amount of storm sewage discharged into small urban rivers during torrential rainfall events may exceed the mean discharges many times [53,54]. It is universally recognized that climate change and the intensification of precipitation are likely to further aggravate this problem in the future [55].

A comparison of the maps presented in Figure 4a,b shows the differences in the range of floods and the amount of flood risk for $Q_{1\%}$ under the conditions of complete destruction of the embankments in the section of the Vistula (Figure 4a) and their local damage (Figure 4b). When the embankments are completely destroyed, the water occupies the area of its former large water bed. In the case of a local embankment break, only part of the water flowing through the channel enters the collapsed area and fills the area located below the water table, located in the area of the break [13].

Flash floods can occur virtually anywhere in Poland—both in steep and flat terrains. Likewise, pluvial urban floods can occur in virtually any town. They are invariably surprising events, as convective rains cannot be forecast early enough.

The risk of urban pluvial (rainfall) floods has been increasing in Poland. This is documented by the Urban Adaptation Plans to Climate Change [56], implemented in 100 large Polish cities, where this threat was identified as significant. As part of the review of the preliminary flood risk assessment [57], for the purposes of updating the FRMP, a categorization of this risk was performed on the basis of the number of interventions by fire brigades related to inundations, as well as the extent and type of damage (Table 1). Table 1 shows that the most serious problems occurred in Kraków, where three times the threat was classified as disastrous (unlike in other cities, where no disastrous threat was recorded). Kraków is a large city of great historical and cultural importance on the River Vistula and a UNESCO World Heritage site at a significant pluvial and fluvial flood risk [58].

**Table 1.** Number of interventions by fire brigades in 2010–2017 related to inundations and category of threat in 10 large cities in Poland (data from Ref. [57]).

| Town | Number of Interventions by Fire Brigades | Category of Threat | | | | |
|---|---|---|---|---|---|---|
| | | Low | Local | Moderate | Large | Disastrous |
| Elbląg | 178 (100) | 10 (6) | 165 (93) | 1 (0) | 2 (1) | 0 (0) |
| Kalisz | 541 (100) | 17 (3) | 519 (96) | 4 (1) | 1 (0) | 0 (0) |
| Konin | 225 (100) | 6 (3) | 218 (97) | 1 (0) | 0 (0) | 0 (0) |
| Koszalin | 63 (100) | 8 (13) | 55 (87) | 0 (0) | 0 (0) | 0 (0) |
| **Kraków** | **2211 (100)** | **1435 (65)** | **763 (35)** | **8 (0)** | **2 (0)** | **3 (0)** |
| Olsztyn | 229 (100) | 19 (8) | 209 (91) | 1 (0) | 0 (0) | 0 (0) |
| Poznań | 863 (100) | 75 (9) | 787 (91) | 1 (0) | 0 (0) | 0 (0) |
| Rzeszów | 854 (100) | 27 (3) | 824 (96) | 3 (0) | 0 (0) | 0 (0) |
| Warszawa | 3183 (100) | 1043 (33) | 2129 (67) | 10 (0) | 1 (0) | 0 (0) |
| Zielona Góra | 417 (100) | 143 (34) | 272 (65) | 2 (0) | 0 (0) | 0 (0) |

Data for Kraków are presented in bold. Percentages are shown in brackets (rounded to integer values).

The increasing sealing of catchments aggravates the problem of violent rainwater runoffs with large volumes [59]. The design capacity of stormwater drainage systems is too small in comparison with the increasing frequency and intensity of torrential rainfalls. In many Polish cities where river valleys receiving stormwater are narrowed by embankments or city boulevards, it is not possible to discharge stormwater into the river during a river flood. An example is Kraków, where in the very center of the city, the valley of the River Vistula has been narrowed to 350 m, which causes technical problems related to maintaining

the stability of the channel and its low boulevards under high flow conditions (100-year discharge and above). Damage to the riverbed and boulevards, resulting in serious changes in the morphology of the Vistula riverbed, occurred in Kraków during the flood in 2010 [41]. In 2015, the average level of sealing of Kraków's surface reached nearly 25%, and in accordance with the city's development plans, a further increase in the level of surface sealing of 10% is projected [13]. In light of the development trends of Polish cities in recent years, the risk of precipitation and urban flooding can be regarded as one of the main threats facing 80% of Polish cities with the number of inhabitants greater than 90,000 [56]. In effect, there are losses related to flash floods, which have to be sustained by local governments, businesses and affected citizens, possibly with the assistance of insurance. When comparing the currently applicable FRMPs and their updates for 2022–2027, a lack of effective assessment of the sources and effects of this hazard can be noted, as well as a lack of systemic solutions aimed at coping with flash floods and urban floods. Poland lacks key legal and planning solutions, which could serve as guidelines for how to reduce the risk of such floods. There is an urgent need to develop a national strategic document, as well as local strategies, and to link spatial planning regulations with the requirements of sustainable urban water management. Investors (e.g., developers) generate risk, which may lead to future compensation claims.

In addition to climate change mitigation measures (aimed at reduction in anthropogenic $CO_2$ emissions to the atmosphere), cities must undertake urgent climate adaptation measures, which will allow them to increase their resilience to weather extremes, which are becoming more extreme in the warming climate.

A blue-green infrastructure, i.e., combined systems of urban greenery and surface waters, is the basis of resilience in the process of adapting cities to climate change. It plays a key role in preventing urban floods and droughts, as well as reducing the severity of heat waves and the urban heat island phenomenon.

The issue of rainwater management in cities and linking this issue to urban policy and spatial planning are becoming of key importance. A blue-green infrastructure (BGI) and the ecosystem services provided by it, which must be reflected in urban planning standards, the cost–benefit analysis of urban investments and social values play a distinct role in adaptation measures and multi-functional urban land management. Unfortunately, BGI still has no legal empowerment in Poland, and the legal protection of trees was loosened in 2022.

The management should include renaturation in order to help trigger the process of environmental and climatic regeneration—that is, linking the regeneration of ecosystems and adaptation to climate change [60]. Krauze and Wagner [61] analyzed existing best practices of nature-based solutions in urban water management, including flood protection, which enable, restore and preserve nature.

### 2.4. Synergy and Trade-Off between Flood and Drought Risk Reduction Measures

There are examples of synergies and conflicts between flood and drought risk reduction measures [62]. The very concept of a multi-purpose dam reservoir illustrates a conflict. For flood risk reduction, it is necessary to keep some pre-defined empty storage volume in order to accommodate a possible flood wave. For drought risk reduction, a flood storage reserve means a lost opportunity for storing more water, which could be very valuable for low-flow augmentation if a hydrological drought develops. For the prevention of drought, a "wet" reservoir would be preferred, while for flood risk reduction, a "dry" reservoir (polder) would be preferred, which would be able to store more flood water. Therefore, what is better for flood risk reduction may not be better for drought risk reduction. However, river restoration techniques, e.g., relocating dikes by increasing their spread and creating polders, can be beneficial for both flood and drought risk reduction [63].

Pawlaczyk et al. [64] described examples of successful renaturation. The restoration of natural flood retention of the Odra River valley in the Wołow commune as a result of the relocation of flood embankments is beneficial for the functioning of the Natura

2000 area called "Łęgi Odrzańskie". Moreover, the project improves flood protection in a narrow section of the embankment area and reduces the risk of dike break and inundation of two villages, as had occurred during the 1997 flood. Another renaturation success story is the optimization of the River Bóbr bed between Bobrów and Wojanów, where a good hydro-morphological state was achieved; the unit power of the streamflow was reduced; and bank stability was ensured. By enriching the landscape with morphological elements favoring biodiversity, the ability to slow down the water flow was restored. The project was carried out through the purchase of land, where meadows and fallow lands were transformed into valuable ecosystems. A significant part of the area, which was drained and dried up years ago, is now protected as the Natura 2000 area called "Dolina Łachy". Good maintenance practices lead to a delay in runoff, e.g., via increasing streambed roughness. The maintenance method should not obstruct the safe spreading of flood waters in the valley.

It is necessary to enhance natural channel retention by improving valley retention (by restoring or increasing the natural floodplain via liquidating or relocating embankments), retention in the catchment area (by restoring wetlands), protection of environmental zones and watercourses in the natural form—including rebuilding their retention by slowing down the outflow, restoring oxbow lakes and enhancing connectivity between the stream and floodplain—and inhibiting the outflow from drainage networks.

The basis for mitigating drought and its effects, as well as buffering floods, recognized at the European Union scale are ecosystem-based solutions and germane notions of blue-green solutions (BGSs) and nature-based solutions (NBSs), which allow for effective water storage in the landscape (see Natural Water Retention Measures, http://nwrm.eu/ (accessed on 8 August 2023)). The blue-green infrastructure (BGI), understood as natural, semi-natural and artificial aquatic and terrestrial ecosystems, is an important element of water and natural hazard management and of adaptation to climate change, particularly in cities. The BGI can provide ecosystem services, which include, among others, the restoration of water cycling processes in urban landscape, thus mitigating floods, supporting the functioning of stormwater drainage systems, as well as reducing drought and urban heat island effects and curbing the costs associated with the occurrence of disturbances. The functioning of BGI in cities is advantageous in the Anthropocene, i.e., the era of intense anthropopressure. This may contribute to reducing the ability of ecosystems to provide their services. It is necessary to develop a coherent systemic approach to the management of rainwater, green areas and water ecosystems in cities with the aim of an integrated flood and drought risk reduction. An example of good practice at the national level is the document of the new National Urban Policy 2030 [65] in Poland (adopted in 2022), which addresses climate and environmental challenges by creating "green cities". At the local level, these recommendations are implemented through water and environmental urban policies (e.g., in the city of Gdansk) and documents from the Study of Conditions and Directions of Spatial Development of Communes. The new study for the city of Poznan, adopted in 2022, is a good example of treating the blue-green network as a priority. According to some opinions, BGI should be treated as a critical infrastructure because the security of cities depends on its effectiveness. Unlike technical critical infrastructure, BGI is dispersed and self-sustaining, so it can be protected more easily. Unfortunately, these studies are not legally binding. Legally enforceable planning acts are local spatial development plans (LSDPs), but according to the OnGeo.pl database (https://resources.ongeo.pl/dokumenty/mpzp/dane-pokrycie-mpzp-publ.zip (accessed on 8 August 2023)), LSDP coverage of the area of Poland is on average only about 32.6%, and there are also voivodeships where the coverage is below 10% (e.g., 7.7% in the Kuyavian-Pomeranian voivodeship) and cities where less than 20% of the area is covered by LSDPs (e.g., Rzeszów: 15.5%; Kielce: 17.8%). In such cities, environmentally valuable areas are often omitted and developed in violation of the study documents on the basis of development conditions issued in the mode of administrative decisions.

Water retention—storing water when abundant and releasing it when scarce—is an essential measure for both flood and drought risk reduction. The efficiency of different types of flood storage systems should be considered in the context of their effect on reducing the volume of flood runoff. Forested areas and other green areas created, as well as small-scale storage capacity or polders as integrated solutions, can reduce the risk of a 5-year or 10-year flood. On the other hand, the reduction in the risk of floods with a high recurrence period (e.g., via protection against a 100-year flood) requires the restoration of river spaces (i.e., "room for the river") and providing a large flood storage capacity in reservoirs. To lower the flood and drought risk, it is necessary to increase the storage capacity of different types—both natural and artificial. Water retention in the landscape, in the river valley or soil retention (based on the understanding that an increase in soil organic matter results in enhanced water storage) should also be included in these plans. It is important to consider the retention capacity of groundwater aquifers and the possibility of their recharge by abundant flood waters.

Appropriate linkage of retention measures must be adapted to actual hydrological, geological and environmental conditions, as well as to existing and planned infrastructure in a river basin. This also requires their efficiency to be monitored within large-scale systems—local to regional—and their adaptation to land use and development plans. Single structures and measures, particularly small ones and unrelated to other measures, are generally insufficient and ineffective.

## 3. Results—Challenges in Flood Risk Management Policy

This section contains the results of authors' assessments of the weaknesses of flood risk management policy in Poland and their proposals for measures, which, if undertaken, could considerably reduce flood risk.

Controlled flood retention in storage reservoirs accounts for only 10–40% of the total capacity of water reservoirs in Poland (on average, it does not exceed 25%). This number includes dry flood reservoirs and river polders. For an effective reduction in flood risk, a significant increase in basin, channel and local storage is necessary, especially in the Vistula River Basin. However, the implementation of this task has not been effective so far.

In general, in both the Vistula and Odra River Basins, the level of implementation of plans for flood risk mitigation measures for the first planning cycle of the FRMP (2016–2021) was very low. For the first time, starting with analysis of the needs and intended measures in the second planning cycle (2022–2027), an assessment of the degree of implementation of intended measures contained in the plans for 2016–2021 was made. It indicated a very low efficiency of the implementation of plans [37,38]. In the Odra River Basin, 300 measures were planned. In reality, only 19 activities (6.3%) were implemented; 48 activities have been initiated and need to be continued in the next planning cycle, while 159 measures were abandoned. Furthermore, the implementation of 25 additional flood risk reduction measures (not included in the FRMP) was undertaken, 10 of which have been fully accomplished. In the Vistula Basin, in the first planning cycle of the FRMP, 1680 activities were planned. In reality, only 44 actions (2.6%) were implemented; 207 activities have been launched and need to be continued in the next planning cycle, while 1137 activities were abandoned. Furthermore, the implementation of 31 additional flood risk reduction measures (not included in the FRMP) was undertaken, 17 of which have been fully accomplished (https://www.gov.pl/web/infrastruktura/rozporzadzenia-przyjmujace-plany-zarzadzania-ryzykiem-powodziowym-podpisane-przez-ministra-infrastruktury (accessed on 8 August 2023)). The following very small portions of the plan have been implemented in the Vistula River Basin: 0.1% of the planned river polders, 2.5% of the reconstruction of valley retention and 0% of reservoir flood retention [38]. This state of the art can be interpreted as a result of the introduction of the water management reform (amendment of the Water Law in 2017–2018) and the acute lack of adequate financial resources. However, regardless of the reasons, this state of affairs not only made it necessary to intensify works in the years 2022–2027 but also caused a domino effect. This refers, for instance, to linking

the changes in spatial planning and recommendations for rainwater management with preventive measures for reducing the impact of development and the effects of climate change on the risk of precipitation and urban flooding in urban areas.

The draft FRMP [37,38] for the second period (2022–2027) (called uFRMP, i.e., the updated FRMP) is ambitious in terms of the capacity to store flood waters but not so ambitious with regard to the increase in the area of river valleys (171 km$^2$, i.e., barely 1.4% of flood-prone areas) [38].

In the authors' opinion, there are several major deficiencies in the existing FRMPs in Poland, as listed below:

(i)    The plans contain too many specific measures, frequently small ones. Many of them come across as inapt and economically ineffective solutions. In this respect, it is important to follow an integrated approach to flood protection and protection of aquatic ecosystems, as well as to include measurability of the effects of FRMP implementation. The measurability may apply to the division of effects into limiting existing threats, preventing new threats and limiting negative effects of flooding.

(ii)   The methodology for preparing FRMPs [66] provides that the measures proposed for inclusion in the action plan should meet specific conditions. They should be adequate to the needs and objectives of flood risk reduction, being well thought out (determination of the location and parameters), well prepared (determination of the implementing agencies), as well as feasible for implementation (with guaranteed financing). They should also satisfactorily meet the economic efficiency criterion. Unfortunately, in many cases, the existing FRMPs do not meet one or more of these conditions. A significant part of the measures are not applicable to the problem areas characterized by the highest flood risk, while at the same time, less valuable areas are not sacrificed for the benefit of more valuable ones. Additionally, for many measures, the benefits that their implementation should bring are not defined. An important weakness of this methodology is also the lack of quantification of the impact of planned new water storage capacities on the flood risk downstream, taking into account flows from downstream tributaries.

(iii)  Finally, the FRMPs include many measures, which do not have much to do directly with flood risk reduction but are often linked to other objectives, such as improvement in inland navigation conditions, which has been vigorously promoted by the political group ruling the country since 2015.

These reservations and comments are also supported by expert opinions prepared for the needs of FRMPs consultations in the short period intended for this purpose [48]. They are also a result of earlier conceptual and planning studies at the regional scale, concerning in particular the basins of the Upper Vistula [13,35] and the Upper and Middle Odra [33]. Recommendations in this regard primarily concern the multi-scale assessment of the effects of flood prevention and flood risk reduction measures in cities, e.g., in Opole, Wrocław, Kraków, Sandomierz and many others. They must take into account their effects on a regional scale.

In the opinion of the present authors, considering the following major challenges could improve the situation:

(i)    When preparing FRMPs, it is important to control the impact of the implementation of measures on the achievement of the assumed goals, given the lack of continuity of databases, analyses and evaluations, and especially of economic efficiency.

(ii)   More nature-based solutions (NBSs) should be included, such as spreading of the flood embankments away from the river channel (e.g., on the River Odra) or use of oxbow lakes as part of polder retention, to effectively increase the storage capacity of the river valley. This applies primarily to the River Vistula, where the multi-channel system was not preserved in the process of regulation, and the high water bed was narrowed to a width corresponding to only 10–50% of the width of the natural valley [35].

In the authors' opinion, the following actions are urgently needed in the domain of urban flood risk reduction (relevant to Section 2.3):

(i)    It is necessary to develop strategies and local rainwater management plans not only in areas prone to flash and urban floods but in entire urban catchments in order to strengthen the preventive measures, as well as the economic tools and recommendations for their implementation. Good exemplary practices include the "Action plan to improve flood protection and drainage in the City of Krakow", dated 2016, and the "Gdańsk Water Policy", dated 2018. It is clear that this type of local approach must be adapted to local conditions. However, there are no general rules and requirements for these types of documents and their implementation.

(ii)   It is necessary to change the design principles of stormwater drainage systems based on a revised approach to heavy rainfall statistics, which are changing with climate change and the increasing levels of catchment sealing. These systems should also take into account the retention capacity of small urban watercourses and blue-green infrastructure. Such methods and tools have been developed in Poland as part of the PANDA project and are now widespread and accepted in a wide range of planning and design practices (https://retencja.pl/ (accessed on 8 August 2023). They should be recognized as official recommendations, reflected in standards.

(iii)  It is recommended to change the approach to land use planning and replace the principle of fast drainage ("from rain to drain") through development of a "sponge city" [67,68] and decentralized rainwater runoff management based on the "source–pathway–receptor" approach [69,70], which incorporates the following measures:

- "at the source"—increase in the storage capacity, infiltration and use of rainwater where it falls across entire catchments in built-up areas, including private and public properties (it is necessary to urgently amend the Spatial Planning and Development Act and Building Law, as well as the provisions in local strategic and planning documents in Poland) and road infrastructure, allowing for temporary flooding of low-lying areas (short-term water storage should be incorporated into multi-functional land management, as well as land and infrastructure development), as well as the requirement for hydraulic neutrality (an unchanged surface runoff rate before and after an investment) of new private and public investments;

- "on the path"—departure from urban drainage systems in favor of retention systems; modeling and upgrading of underground networks and relieving these networks by connecting them to systems of open drainage ditches, canals, small watercourses and storage reservoirs, which would improve the flexibility of the system, the retention capacity and the possibility to control the flow of water based on stormwater runoff management plans;

- "in the receptor"—reduction in investments in areas at risk of local flooding in favor of increased storage space for water and the possibility to pre-treat stormwater runoff (e.g., via buffer parks) but also improvement in connectivity, biodiversity and recreation conditions. It is necessary to develop standards for land development and management in areas at risk of flooding, in combination with a flood insurance system. It is also recommended that the pollution load should not surpass the ability to maintain a good condition of the water bodies. In the coastal areas of rivers, which receive stormwater discharges, planning documents (within the range of the backwater impact of rivers) should take into account the conditions for the passage of river flood [50].

The new document for conducting urban policy, i.e., the National Urban Policy 2030 (https://www.gov.pl/web/funds-regional-policy/national-urban-policy (accessed on 8 August 2023)) (NUP 2030) adopted by the Polish government on 14 June 2022, aims to build the conditions for strengthening the capacity of cities and their functional areas for sustainable development, as well as building resilience to climate change and improving residents' quality of life. The key challenge for the issues raised in the article is mitigating

the negative effects of climate change in cities, which emphasizes the need to implement the following solutions:

(i)     Introduction of the standard of protection and shaping greenery in investment processes;
(ii)    Legal empowerment of BGI;
(iii)   Managing water resources in the catchment system;
(iv)    Financial, legislative and organizational mechanisms benefiting an increase in natural retention;
(v)     Counteracting urban floods and droughts and the effects thereof with legislative changes;
(vi)    Introduction of the urban BGI management plan as the implementation of the recommendation to draw up a "greening plan" included in the EU Strategy for Biodiversity 2030.

In order for the NUP 2030 policy not to remain merely a declaration, effective planning and financial tools are needed. They should be included in the Spatial Planning and Development Act and the Construction Law, as well as in the Nature Conservation Act in Poland.

In the authors' opinion, to achieve a synergy of flood and drought risk reduction measures, several activities are potentially promising, as presented in Table 2 (relevant to Section 2.4).

**Table 2.** Recommended activities for achieving synergy of flood and drought risk reduction.

| # | Activity (Institutional Level of Implementation) | Main Metrics | Explanation |
|---|---|---|---|
| i | Develop and implement integrated, long-term flood and drought risk management policy in Poland (at all planning levels) | Reduction in the frequency of floods and droughts in relation to the frequency and spatial risk of precipitation excess and deficiency. | Synergy should be the basis for measures taken to mitigate the combined flood and drought risk, with cost optimization. Separate treatment of flood risk and drought risk in planning may lead to solutions, which "do not know about each other" and can be in conflict with each other (cf., Section 2.4). A solution aimed at flood reduction may create disbenefits for drought risk reduction and vice versa. By seeking a compromise, we may not solve any of these problems. |
| ii | Include the "National Surface Water Renatu-ration Programme" in the River Basin Management Plans (at the national level in the planning system and the regional or local level in the implementation) | Assessment of the ecosystem functionality of rivers in sections of length, ensuring a real balance of its biological and economic functions. | River Basin Management Plans are basic documents implementing the EU Water Framework Directive (WFD) [3], which are under preparation in Poland. Their implementation should take into account the requirements of the ecosystem functionality of the rivers they concern. |
| iii | Prioritize drought and flood risk reduction measures (at all planning levels) | Cost–benefit ratio (taking into account environmental, social and economic benefits), measured on a spatial scale adequate for the extent of expected impacts and a time scale appropriate for balancing the costs and benefits. | Introduce an efficient system for prioritization of drought and flood risk reduction measures, a mechanism for their selection and the conditions for accepting them for implementation in order to enhance the efficiency and cost effectiveness of solutions. Only measures with an acceptable cost/benefit ratio should be included in planning analyses. |

**Table 2.** *Cont.*

| # | Activity (Institutional Level of Implementation) | Main Metrics | Explanation |
|---|---|---|---|
| iv | Upgrade urban planning documents (at regional and local levels) | Environmental performance measures included in the updated plans. | Include provisions in urban planning documents, which, apart from maintaining a minimum proportion of biologically active areas, will also ensure better conditions for the functioning of natural systems and support rainwater management in cities, including the following: preservation of the continuity of the natural system, restoration of species diversity and access to water by connecting green spaces with rainwater retention systems. |
| v | Jointly treat the blue-green infrastructure in cities and their surroundings (at local level) | Measures of the impact of land use on water management and ecological efficiency. | Necessary cooperation of local governments and various entities responsible for water management and planning in order to ensure connectivity of the blue and green infrastructure in cities and their surroundings and in the region, including limiting unfavorable interactions between urban and non-urban areas, e.g., for flood risk, the risk of water deficit and over-exploitation of groundwater. |
| vi | Plans for storage capacity of many types should be prepared for problem areas (at regional and local levels) | Measures of real assessment of the effectiveness of the planned retention at the scale of the area in relation to the expected retention functionality and the level of its effectiveness at the stages of its spatial and temporal implementation. | Improve the planning and implementation of water retention measures geared toward flood and drought risk reduction. Plans for the development of multi-type storage capacity should be prepared for problem areas, including the determination of monitoring cross-sections to balance the effects and assess the phased implementation of these plans. Such plans must be based on study documents, which take into account hydrological, hydrogeological and environmental conditions. They should define the development effort and determine the impact of climate change. |

**#** Self-explanatory numbering.

## 4. Discussion

The present section sketches the way ahead for flood risk management in Poland within the timeframe of the implementation of plans for 2022–2027 and the next regular update of plans for 2028–2033.

Flood risk management plans (FRMPs) exist in Poland for the first planning period (2016–2021) and for the second planning period (2022–2027). As demonstrated in Section 3 of this paper, these plans—obligatory under EU Floods Directive—show significant weaknesses. The implementation of the planned actions in the first planning period (2016–2021) turned out to be very low—of the order of just a few per cent. Only 6.3% of the number of planed actions were implemented for the Odra River Basin and 2.6% for the Vistula River Basin (see Section 3 of the present paper). The flood risk management plans (FRMPs) were not correlated at all, with scanty financial resources available for implementation. In order

to increase the effectiveness of the implementation of plans in the years 2022–2027, as well as to strengthen the systemic approach in planning flood risk reduction in the next (third) planning period (2028–2033), the following activities might help:

(i)　Preparation of a catalog of financial, legal and governance (competence, institutions) instruments necessary for an effective implementation of the planned flood risk reduction measures. The catalog of these instruments could take into account the areal integration of multi-type activities. In this way, effective and measurable areal effectiveness of the phased implementation of the planned flood risk reduction projects can be enhanced.

(ii)　In the years 2022–2025, for the purpose of updating the plans for 2028–2033, it is recommended to develop a flood risk assessment system in the field of pluvial floods, dominating in cities. Pluvial floods have not been included in the update of flood risk management plans for 2022–2027. This type of flooding may dominate in the future, as short-duration rainfall is more likely to exhibit greater increases [71]. Moreover, some heat waves, which are becoming more intense in cities, may culminate in heavy rainfall [72]. In the national dimension, it is urgent to identify the areas of potential threat and its sources, as well as the negative effects of flooding, and to develop a long-term forecasting system in order to reduce pluvial flood risk. Currently, and in the near future, cities are deprived of systemic support in this regard, especially under the conditions of simultaneous pluvial and fluvial flooding.

(iii)　In the update of the plans for 2028–2033, measures for reducing flood risk might be grouped within the boundaries of areas, enabling a realistic assessment of flood risk reduction as a result of the systemic implementation of projects reducing this risk. This applies in particular to areas prone to floods of various types. This could make it possible to move away from lists of separate (as well as difficult to grasp) and sometimes minor interventions in favor of a systemic spatially integrated grouping of them. This could also allow for effective prioritization of activities and staging of projects.

(iv)　A realistic approach for selecting structural measures might be possible, taking into account the real possibility of financing, which would concentrate the FRMPs for 2028–2033 on priority measures for flood protection in the Vistula and the Odra River Basins.

The implementation of the above measures would also require the development of national planning documents:

(i)　The integration of spatial planning with water management planning, including flood risk reduction, should be enhanced at all management levels, leading to long-awaited legislative amendments based on documented experience;

(ii)　The effectiveness of the implementation of the new National Urban Policy 2030, which recommends, among others, the creation of green and resilient cities, would require significant changes to the Spatial Planning and Development Act and the Construction Law to ensure legal empowerment for the protection and development of blue-green infrastructure;

(iii)　Plans for the development of multi-type water storage capacity in river basins should be prepared, taking into account the current conditions and development efforts and the impact of climate change, as well as land use and land cover changes, on flood hazard.

As far as national strategic actions are concerned, it is necessary to develop a long-term water management strategy based on a participative system, i.e., with the participation of expert and academic communities, water users and NGOs. This approach should include a coherent flood risk reduction strategy.

The enhancement of competence potential in water management requires updating the state of knowledge and skills, primarily through the following:

(i)     Strategic and regional studies conducted by the scientific and academic community, in cooperation with the administration and consultancy companies, seeking effective solutions in flood risk reduction;

(ii)    Continuous training of specialist personnel who deal with flood risk reduction, according to a well-thought-out system of life-long learning, including review of the curricula of higher education institutions.

For centuries, technical (hydraulic) structures had been the principal flood risk reduction measure in Polish lands [73]. The present priorities of the Polish government related to flood risk management typically focus on large infrastructure solutions, but opportunities for large-scale solutions in the country are limited due to topography, environmental and social concerns, as well as high costs. The interventions are economically promising if the costs of the investments are clearly lower than the recurrent costs of losses if nothing is done. It is important that each investment has net benefits relative to the existing alternatives.

A portfolio of integrated solutions is needed in Poland, which can be bundled into regional actions aimed at flood risk reduction. The components of this portfolio could be integrated (addressing multiple objectives), economically and socially feasible, adaptive to uncertainty, robust, as well as evidence based.

The authors recommend a range of necessary and urgent actions to be taken by decision makers and politicians, who are responsible for the formulation of flood risk management actions in the spirit of the EU Floods Directive and their implementation. The publication of a roster of expert recommendations in a peer-reviewed open-access journal of international standing is likely to be an effective vehicle for drawing the attention of the authorities in Poland.

This paper, advocating for integrated flood risk reduction, is likely to be of interest to the international readership, particularly in the countries of central and eastern Europe, where similar challenges in the planning of flood risk management—obligatory through the EU Floods Directive—and implementation of plans may also feature.

## 5. Conclusions

Floods are the main natural disasters in Poland. Multiple fluvial, pluvial, snowmelt, ice-jam and coastal floods have been recorded in the country in recent decades. The destructive flood of July 1997 in the Odra River Basin caused particularly high human and material damage. The number of fatalities in the Polish part of the basin was 55, and the material damage was estimated at PLN 12.5 billion (USD 3.6 billion), which accounted for about 2.4% of the Polish GDP. Pluvial floods are on the rise in the changing climate, while an increase in the area of sealed urbanized surfaces exacerbates the severity of urban inundations.

This paper provides an overview of the state of the art in flood risk management in Poland, embracing the current legislation, regulations, plans, strategies and measures undertaken.

The challenges for flood risk reduction in Poland include the need to adapt water management to the progressing climate change and spatial and temporal changes in precipitation distribution. Serious challenges are related to the necessity for organizational and legal changes enabling the integration of water management with the protection and restoration of the environment of river valleys and wetlands, as well as more sustainable land use in urban and rural areas. These changes result from the need for real and effective integration of planning in the field of water management and flood risk management at the national, regional and local levels.

The integration in question requires skills and consistency in respecting the conditions, which should be subject to a selection of measures, which reduce flood hazard and flood risk due to the existing and future environmental conditions affecting the ecological status of water ecosystems. Taking into account the development plans and the effects of climate change, the procedures applied in this area, and consequently, in planning activities at appropriate spatial and time scales, should use the full scope of the driver–pressure–state–impact–response (DPSIR) analysis.

The existing weaknesses in the flood risk management plan in Poland for the first planning period (2016–2021) and for the second planning period (2022–2027) are discussed, and the paths for improvement are proposed. The level of implementation of the plans in the former period was very low. Furthermore, many planned measures did not have much to do with flood risk reduction but were often linked to other objectives, in particular to inland navigation, which seemed to be an utmost priority in Poland. The existing plans contain numerous small measures, which come across as inapt and economically ineffective solutions. The existing priorities typically focus on large infrastructure solutions, while nature-based solutions, which are really urgently needed, are left behind.

**Author Contributions:** Conceptualization, Z.W.K., A.J.-S., E.N., I.P. and J.Z.; methodology, Z.W.K., A.J.-S. and E.N.; investigation, Z.W.K., A.J.-S., E.N., I.P. and J.Z.; writing of the first draft, Z.W.K.; writing—review and editing, Z.W.K., A.J.-S., E.N., I.P. and J.Z.; visualization, E.N., A.J.-S. and Z.W.K.; supervision, Z.W.K. All authors have read and agreed to the published version of the manuscript.

**Funding:** I.P. was supported by the National Science Centre of Poland (project number 2018/31/B/HS4/03223).

**Acknowledgments:** The authors gratefully acknowledge Jerzy Hausner and five anonymous reviewers for their useful and constructive remarks, which helped us improve this paper.

**Conflicts of Interest:** The authors declare no conflict of interest.

## Abbreviations

| | |
|---|---|
| BGI | Blue-Green Infrastructure |
| BGS | Blue-Green Solutions |
| EU | European Union |
| FD | EU Floods Directive (Directive 2007/60/EC) |
| FRMPs | Flood Risk Management Plans |
| KZGW | National Water Management Authority in the structure of theNational Water Holding-Polish Waters (PGW-WP) |
| NBSs | Nature-Based Solutions |
| NGOs | Non-Government Organizations |
| RBMPs | River Basin Management Plans (RBMPs) |
| PGW-WP | National Water Holding-Polish Waters |
| uFRMPs | Update of the Flood Risk Management Plans in the third planning cycle (2022–2027) |
| uRBMPs | Updated River Basin Management Plans (RBMPs) |
| WFD | EU Water Framework Directive (Directive 2000/60/EC) |

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
