# Peer review of "Challenges for Flood Risk Reduction in Poland’s Changing Climate"

_water, doi:10.3390/w15162912_

Round 1
Reviewer 1 Report (Previous Reviewer 1)
The authors of the paper took up a very important problem related to contemporary climate change, namely the problem of flooding in Poland. The work is generally well written and I have no major comments. The work has been corrected and supplemented in relation to the version I reviewed. Several literature items have been added, also regarding floods that have occurred in historical times. The authors also separated the discussion from the conclusions, which makes the structure of the manuscript clearer. Only the drawings need improvement: that is, you need to add line scales where they are missing. I would also advise supplementing the literature review with two papers: one concerns hydrographic conditions in the Vistula valley in historical times and floods in the past:
Czaja, Jakub. "Hydrological effects of the hydraulic structures constructed in the valley of the River Little Vistula in Poland from the mid-18th century to the present" Environmental & Socio-economic Studies, vol.5, no.1, 3917, pp.25-36 . https://doi.org/10.1515/environ-2017-0003
and geospatial analysis of the course of flash floods in urban areas:
Bandi, Aneesha Satya, Meshapam, Shashi and Deva, Pratap. "A geospatial approach to flash flood hazard mapping in the city of Warangal, Telangana, India" Environmental & Socio-economic Studies, vol.7, no.3, 3919, pp.1-13. https://doi.org/10.2478/environ-2019-0013
Good luck
Author Response
Authors’ responses to reviewer #1
Comment:
The authors of the paper took up a very important problem related to contemporary climate change, namely the problem of flooding in Poland. The work is generally well written and I have no major comments. The work has been corrected and supplemented in relation to the version I reviewed. Several literature items have been added, also regarding floods that have occurred in historical times. The authors also separated the discussion from the conclusions, which makes the structure of the manuscript clearer. Only the drawings need improvement: that is, you need to add line scales where they are missing. I would also advise supplementing the literature review with two papers: one concerns hydrographic conditions in the Vistula valley in historical times and floods in the past:
Czaja, Jakub. "Hydrological effects of the hydraulic structures constructed in the valley of the River Little Vistula in Poland from the mid-18th century to the present" Environmental & Socio-economic Studies, vol.5, no.1, 3917, pp.25-36 . https://doi.org/10.1515/environ-2017-0003
Response:
This paper is now added to the list of references
Comment:
and geospatial analysis of the course of flash floods in urban areas:
Bandi, Aneesha Satya, Meshapam, Shashi and Deva, Pratap. "A geospatial approach to flash flood hazard mapping in the city of Warangal, Telangana, India" Environmental & Socio-economic Studies, vol.7, no.3, 3919, pp.1-13. https://doi.org/10.2478/environ-2019-0013
Response:
It is likely that there exist thousands of papers that are more relevant and more worthy of citing in our paper. The paper by Bandi et al. is quite technical, addressing a narrow topical area, and of local importance, giving an example of flood hazard mapping in a town in India. If we cited this paper, the readers could be really puzzled – why?. Sorry, we do not add it to the list of references.
Reviewer 2 Report (Previous Reviewer 3)

Author Response
Authors’ responses to reviewer #2
Recommendation
Minor Revision
In general, the article presents a certain degree of complexity, addressing a subject about which it is difficult to find compact and complete information in the administrative documents and basin management plans existing at the level of a country. However, the research is quite descriptive and exploratory, not providing many statistical data regarding certain types of management measures. The authors summarize, in many respects, to offer a critical analysis of insufficiently implemented measures, or even non-existent, specifying what should be done in their place. However, we do not have a quantification of the already existing and functional measures, despite the very pertinent examples given from the two large hydrographic basins, Odra and Vistula. However, at the state-of-the-art level, the article offers a rather good start in identifying weak points and to be remedied in flood management in Poland.
In the new version, the authors took account on the changes suggested by the reviewers and visibly improved the manuscript. There is now a clearer and complete text, with less ambiguity and more details on the framework. There are only a few small adjustments to be made in order to be ready for publishing. Also, please allow your manuscript to be proof read by an English proficient, before submitting the final version.
Response:
The authors are very grateful to reviewer #2 for another careful review and many good advices.
Comment: Lines 33 – 34: Please revise and rephrase in a more impartial / formal style the following phrase – “most Polish people think of water 33 when there is a spectacular problem”.
Response: Done.
Comment: Line 72 : I do not find the term “taxonomy” much appropriate in this context. I would suggest “pattern”, or “behavior”. Or, if changed the whole expression, I would rather use “genetic classification of floods” or “causative classification of floods”, as it is more used in the literature. In the case of the term employed in this paper, “taxonomy” would rather refer to the denomination of some categories of floods, which is not the case. You are, in fact, emphasizing the synoptic/ meteorological conditions of flood events.
Response: Now we have: “genetic classification of floods”
Comment: Line 138 – I suggest to emphasize “was not affected by other large floods that …”
Response: Done.
Comment: Line 146 – add “the” at “in … last”
Response: Done.
Comment: Line 157 – What parameter does “amplitude” refer to?
Response: Amplitude of stage and discharge. Text revised.
Comment: Figure 1 – The symbols are still oversized and, thus, partially. Overlapping. Also, I emphasize one more time the importance of delineating the three main watersheds (Elbe, Odra and Vistla). The scale is missing.
Moreover, it is still unclear how the floods were attributed to the gauging stations. For instance, if a certain flood event is not symbolized on the map, does it mean that it did not occur at all on that river reach, or that it is not considered large? If the latter, I resume the question from the previous review round. When does a flood become large “large”? Please, explain the criteria behind the classification of large flood events symbolized on the map.
Response: In the opinion of co-authors, Figure 1 is a useful orientation scheme as it is. Reviewer #2 stated “I emphasize one more time the importance of delineating the three main watersheds (Elbe, Odra and Vistla).” The Polish portion of the Elbe watershed is very small (0.08% of Poland’s territory) and it is largely a Czech-German river. Two basins of the really large rivers: the Vistula and the Odra are outlined in Fig. 2 (including the bits of the basins beyond the Polish boundaries, as per recommendation of reviewer #4 in the first round). Marking the Vistula and the Odra basins in Fig. 1 would render this figure too complicated and less transparent, hence – in our opinion – unnecessary. Marking of the tiny (less than one pro mille of the Poland’s area) bit of the Elbe catchment does not make any sense. There are further rivers catchment on the Polish territory, apart from the Vistula and the Odra, namely: the Pregola, the Nemunas, the Dniester, the Danube. Furthermore, there are smaller coastal rivers in the north. We do not agree with changes proposed to Fig. 1 by reviewer #2, so we considered elimination of this Figure but this would be really unfortunate, because we trust that it is very informative – showing a lot of information on a small orientation scheme. In order to solve the problem we added information that floods of regional extent in Poland occurred in the catchments of the Vistula, the Odra, and the coastal rivers.
The idea of the map in Fig. 1 dates back to late Andrzej Dobrowolski, a co-author of the chapter on floods in Poland in the book edited by Kundzewicz in 2012, who introduced distinction between regional and local floods depending on the inundated area. Local floods are defined as those with up to flooded area of a few hundred km2, while floods of regional extent have greater inundated area (say from 500 km2 upwards). Hence, the term “large” refers to the area, but rather than explaining this in the text, we decided to take this word out. Some highly destructive floods in Poland (like the 1998 flood in Polanica Zdrój, with fatalities) were local, not regional, so that they are not included on the map. We are not going to defend the word “large” of rather subjective connotation. The 1998 flood in Polanica Zdrój could be qualified as large (causing fatalities) but not of regional extent.
Comment: Line 219 – 220: Unclear formulation in “increasing trends… were strongest”. The comparative degree of “strong” may be wrongly employed. Please rephrase whether it is “the strongest” or “stronger than before/ another reference period”.
Response: Authors thank reviewer #1 for indicating this weakness. Indeed, we fixed the point by replacing “strongest” by “strong”.
Comment: Lines 226 – 227: No need for comma in “, spatially varied,”
Response: Now, we have “spatially-varied”, without two commas.
Comment: Line 268 – Add comma in “, implemented in 1998 – 2006,”. The same on the lines 271 and 274.
Response: Done.
Comment: Line 278 – I do not see the logic of using the conjunction “but”, as the improvements on Odra river do not constitute a contrasting example.
Response: Re-worded. Indeed, the improvements on the River Odra do constitute a contrasting example with those on the River Vistula. They are considerably more successful on the Odra than on the Vistula.
Comment: Line 288 – 289 : Add the comma in “, and of the Odra River 288 Basin in particular,”.
Response: Done.
Comment: Line 289: Replacement of “now” with “nowadays”.
Response: Done.
Comment: Line 490 : I would suggest to replace “sufficiently early” with “early enough”.
Response: Done.
Comment: Line 649: Odd syntax formulation in “There exist options that BGI should be…”. Please, rephrase it.
Response: It is not options but opinions. The usage of the word “options” would make no sense, while the usage of the word “opinions” makes sense: “There exist opinions that BGI should be treated as critical infrastructure, because the security of cities depends on its effectiveness.
Comment: Below the line 679, make sure the format of the manuscript allows you to include footnotes.
Response: We accommodated the contents of the footnote in the text. Just to be on the safe side if footnotes are not allowed.
Comment: Line 788 – The use of “read” at the end of the phrase does not have sense. Please revise it.
Response: Re-phrased.
Comment: Lines 789 – 792 : Although the rightness of your assertions is there, I would suggest to use a milder approach in this paragraph, when explaining the “wish list” of the FRMP measures, instead of concrete actions to take.
Response: Milder wording introduced.
Comment: Lines 806 – 907: Add comma before “dated” and after each year.
Response: Done.
Comment: Line 805: What have cities been doing, in “Cities have been doing this…”?
Response: Re-worded.
Comment: Line 850: Needs a reference.
Response: Done.
Comment: Line 855: Key challenges for what ?
Response: Re-worded.
Comment: Lines 913 – 914: The FRMP of the first period, before 2016 does not exist in Poland?
Response: 5. The EU Floods Directive states in Chapter IV, Article 7, Item 5 that “Member States shall ensure that flood risk management plans are completed and published by 22 December 2015.” This referred to the first FRMP period (2016-2021). In Poland as well as in other EU countries, there was no EU-dictated obligation to prepare a FRMP for the years before 2016.
Comment: Line 915: “many items” refer to what aspects of the FRMPs? I would particularize it with something like “several chapters related to../ encompassing problems such as…”.
Response: We re-worded the phrase. Now the vague and unnecessary term “many items” is deleted.
Comment: Line 917: “turned out to be very low – of the order of just a few per cent” – please, be more specific.
Response: Specification was given earlier in Chapter 3, third paragraph. However, for the convenience of the readership (but at the cost of redundancy), we summarize the specific numbers: 6.3% for the River Odra and 2.6% for the River Vistula also here.
Comment: Lines 925 – 927: Be more concise. It is difficult to follow the phrase and the meaning may be lost at some point.
Response: Re-worded now, for better understanding.
Comment: Line 937: What do you mean by “the principles of reducing this risk”?
Response: Re-worded, to avoid ambiguity.
Line 941: Two characters space before ”within”.
Response: Done here and in three other instances in the paper where two space characters were replaced by one space character.
Comment: Line 1010: I would keep “is” instead of “are”. The agreement is made with the portfolio (singlular).
Response: Done.
Comment: Line 1020: Comma after “reduction”.
Response: Done here and in two further places in the text.
Comment: Lines 1050 – 1052: Unfinished phrase in “not only… and thus”. It lacks the counter situation, introduced by “But also”.
Response: Re-worded.
Comment: Line 1073 : Suggestion – replace “badly” with “urgently”.
Response: Done.
Reviewer 3 Report (Previous Reviewer 4)
The new version of the article "Challenges for flood risk reduction in Poland’s changing climate" underwent significant changes carried out by the authors. I recommend your approval.
Author Response
Thanks to this reviewer for recommendations of approval
This manuscript is a resubmission of an earlier submission. The following is a list of the peer review reports and author responses from that submission.
Round 1
Reviewer 1 Report
The authors of the paper took up a very important problem related to contemporary climate change, namely the problem of flooding in Poland. The work is generally well written and I have no major comments. However, several disadvantages should be eliminated:
1. the paper was not prepared in accordance with the guidelines of the journal (method of citation)
2. The lack of conclusions should be supplemented
3. The explanations in the first figure should be placed under the map, the explanations should be given a title (it is not very clear what the date range above the map means)
4. In figure 4, there are no explanations regarding, for example, land use
5. The work concerns the floods that occurred in Poland in the recent past, it is also an attempt at forecasting, but perhaps it would be worth referring to the work concerning the floods that occurred in this country earlier, e.g.: Czaja, Stanisław W., Machowski, Robert and Rzętała, Mariusz. "Floods in the Upper Part of Vistula and Odra River Basins in the 19th and 20th Centuries / Floods in the Upper Part of the Vistula and Odra River Basins in the 19th and 20th Centuries" Chemistry-Didactics-Ecology-Metrology, vol.19, no.1-2 , 2015, pp.127-134. https://doi.org/10.1515/cdem-2014-0012 ?
Reviewer 2 Report
This paper is not suitable for publication in the Water journal. I would categorize it as an average professional paper rather than an original scientific paper or scientific review paper.
The topic of the paper, flood risk reduction, is very interesting and falls within the scope of the Water journal, however, the paper is not internationally relevant and is incomplete. The scientific merit is very low. There is complete absence of the originality or novel approach in this paper. The methodology of the paper is unclear, the Materials section does not describe the materials or methods used in this paper. The entire paper is exclusively focused on the presentation of the state of flood risk management in Poland and is of local importance. The authors almost never refer to examples of (good or bad) practice of flood risk reduction/management in other European countries. On the contrary, they emphasize that the main results of the work are based on expert opinion and individual views of the co-authors, and do not support their views with an argumentative discussion, which, above all, will be based on data, models, concepts, etc. Almost the entire paper, up to the discussion chapter, is listing positive and negative examples of flood risk management practices in Poland, which is basically a very long introduction to the topic of the paper. Even in this very long introduction, the authors unnecessarily list some data and information, and often do not even cite literature sources (for example, the first paragraphs in chapter 3.1). Furthermore, it should be emphasized that the authors do not cite the literature in the text in accordance with the instructions for the authors, and the quality of the images is very poor, often without legends and accompanying explanations. In the list of literature, the authors most often list Polish authors and literature that was originally written in the Polish language.
Since the authors did not prepare their manuscript in a pre-prepared template in which I could make certain comments in the text that refer to a specific line in the text, I am not able to present specific comments on individual parts of the text here.
Reviewer 4 Report
The article addresses a very important topic for the region and allows for applications in other countries, in the context of climate change. It presents contextualizations with long series of data and proposals that will make it possible to carry out plans to minimize the effects of floods.
The article is well written and structured. I recommend approval as-is, and just one suggestion:
- In Figure 2, present the entire watershed of the Odra River, including areas in other countries. This will allow us to better understand the events that occur in Poland and that are the result of hydrological events that occur in other countries.
Reviewer 5 Report
This article is interesting and well structure. Some improvements are necessary so as to be accepted for publication in the Water journal. The main points are listed below:
In the abstract add epigrammatically some major weaknesses points of the existing 19 Flood Risk Management Plans.
Line 51. Add some comments and references about the need for flood exposure analysis of critical/social infrastructure so as to perform a rational flood risk assessment (https://doi.org/10.3390/hydrology9080145, https://doi.org/10.1016/j.ijdrr.2019.101240. In the last decades the advantages of geospatial technologies and open-access data make feasible the flood exposure analysis at national scales providing useful insights to policymakers and stakeholders,
Line 75 – 83. Avoid the repeated use of expressions like “we refer”, “we point out”, etc. Passive voice is preferable.
Add some targets for future research directions that arise from your analysis
The spatial autocorrelation (eg. Moran’s I) of the flood records with the watersheds (or else) conditions will give sufficient information. Is it feasible to perform such an analysis? If not, may be a target for future research?
